# Effects of Olfactory Stimulation with Aroma Oils on Psychophysiological Responses of Female Adults

**DOI:** 10.3390/ijerph19095196

**Published:** 2022-04-25

**Authors:** Na-Yoon Choi, Yu-Tong Wu, Sin-Ae Park

**Affiliations:** 1Department of Bio and Healing Convergence, Graduate School, Konkuk University, Seoul 05029, Korea; nyoony16@naver.com (N.-Y.C.); aaronical920506@gmail.com (Y.-T.W.); 2Department of Systems Biotechnology, Konkuk University, Seoul 05029, Korea

**Keywords:** electroencephalogram, aroma therapy, essential oil, horticultural therapy

## Abstract

This study investigated the effects of olfactory stimulation with aroma oils on the psychophysiological responses in women. Ten aromatic oils (lavender, rosemary, rose, eucalyptus, jasmine, geranium, chamomile, clary sage, thyme, and peppermint) were used on 23 women aged between 20 and 60 years. They inhaled the scent for 90 s through a glass funnel attached to their lab apron, 10 cm below their nose, while the pump was activated. Electroencephalography, blood pressure, and pulse rate were measured before and during inhalation of the aroma oils. The relative alpha (RA) power spectrums indicating relaxation and resting state of the brain significantly increased when lavender, rosemary, eucalyptus, jasmine, chamomile, clary sage, and thyme oils were inhaled compared to those of before olfactory stimulation. The ratio of alpha to high beta (RAHB), an indicator of brain stability and relaxation, significantly increased when rosemary, jasmine, clary sage, and peppermint oils were inhaled. The relative low beta (RLB) power spectrum, an indicator of brain activity in the absence of stress, significantly increased when stimulated with lavender, rosemary, rose, and geranium scents. Further, systolic blood pressure significantly decreased after introduction of all 10 types of aromatic oils, which indicates stress reduction. Thus, olfactory stimulation with aroma oil had a stabilizing effect on the prefrontal cortex and brain activity and decreased systolic blood pressure.

## 1. Introduction

Recently, rapid social changes and persistent COVID-19 worldwide have caused extreme anxiety and stress. Stress responses are a representation of disease, and manifest as physical, psychological, and mental problems. In addition, as the socially-distanced era began, people responded with increased interest in healthy bodies and minds and an increase in the recognition of aromatherapy as a nature-friendly therapy for well-being, wellness, and healthcare [1].

Aroma oils were first used in ancient Egypt. Since then, they have been used worldwide in the treatment of diseases (antibacterial, antifungal, antiviral, and anti-inflammatory), and to improve conditions such as insomnia, depression, anxiety, and cognitive impairment through active treatments such as inhalation, skin absorption, or ingestion [2,3,4]. Aromatherapy is recognized as a field of alternative medicine with the aim of maintaining human homeostasis, using oil extracted from plant leaves, stems, and roots as the main ingredients [5,6]. In addition, people encounter various fragrances in daily life through olfactory stimulation that can initiate physiological effects on mood, stress, and work ability [7].

According to research, the fragrance component of aroma oil is transmitted to the brain via the olfactory nerve through inhalation. Signals from the brain then travel throughout the central nervous system, including the limbic system and hypothalamus, which control human instincts and emotions [8,9].

Studies have reported that scent-related olfactory stimulation produces immediate changes in physiological parameters such as blood pressure, muscle tone, pupil dilation, skin temperature, pulse rate, and brain activity [10,11,12]. In recent years, many scientific studies have been conducted to investigate the effects of aroma inhalation on human brain function. The psychophysiological changes induced by fragrance inhalation have been evaluated using various methods such as electroencephalography (EEG), near-infrared spectroscopy, and functional magnetic resonance imaging [10,13,14].

After inhaling an essential oil mixture of bergamot, geranium, lavender, and clary sage, the EEG of a middle-aged woman under stress revealed that alpha waves were significantly increased in the left and right prefrontal cortex [15]. In addition, it has been reported that inhalation of rose essential oil inhibited the increase in salivary cortisol levels in healthy college students exposed to stress [16]. Igarashi et al. [17] used near-infrared spectroscopy (NIRS) to reveal that the olfactory stimulation using rose oil and orange oil significantly decreased the concentration of oxyhemoglobin in the left and right prefrontal cortex.

Previous studies reported that the correlation between brain activity through olfactory stimulation and human central nervous system activity can be understood using electroencephalography (EEG) [18]. An EEG study found that the scent of Japanese plum reduced EEG readings associated with negative emotions and memory impairment [19]. In addition, contact with natural plants has been found to enhance alpha and beta wave readouts, which reflect a reduction in mental stress [20]. Brauchli et al. [21] investigated physiological changes in response to olfactory stimulation with phenylethyl alcohol (pleasant) and valeric acid (unpleasant) and suggested that unpleasant odors increase alpha 2 power, leading to cortical inactivation.

In a study measuring EEG changes before and after inhaling *I. helenium* essential oil, showed that inhalation of the essential oil activates brain arousal [22]. Jung et al. [23] found a positive effect of the scent of *Lavandula angustifolia* on brain electrical activity in adult women with sleep disorders.

Additionally, some previous studies have suggested that male and female brains are differentially lateralized with respect to cognitive function. In addition, the resting male and female EEG activity differs in the excitatory dynamics of cortical networks, and sex differences have been found in stimulated and non-stimulated conditions [24,25].

As such, various studies have measured the psychophysiological responses to olfactory stimulation using aroma oil, but the oils used in the studies were limited to one or two types. Additional studies investigating differential effects according to sex are needed.

In this study, olfactory stimulation with 10 aromatic oils was performed on adult women, and the effects of EEG changes in the prefrontal cortex, blood pressure, and pulse rate on psychophysiological stability and brain activity were investigated.

## 2. Materials and Methods

### 2.1. Participants

Twenty-three women aged between 20 and 60 years participated in this study. Participants were recruited using a convenience sampling method. A flyer that included study information was distributed to apartments and libraries in Gwangjin-gu, Seoul, Korea. Participants were recruited according to the selection/exclusion criteria shown in Table 1 so as not to influence other physiological data. Before conducting the experiment, participants were informed of the research contents and precautions, with written informed consent obtained before participation in the research. To collect participants’ demographic information, age, sex, height, weight, and body mass index (ioi 353; Jawon Medical, Gyeongsan, Korea) were recorded. Participants received the equivalent of USD 10 as an incentive to complete the experiment. This study was approved by the Bioethics Committee of Konkuk University (7001355-202103-HR-429).

### 2.2. Experimental Environment

This study was conducted in an experimental space (180 cm × 160 cm) in Konkuk University (Figure 1). To minimize external visual stimulation, white hardboard paper was placed before the desk with ivory-colored curtains installed on either side. The environmental conditions of the experiment space were as follows: temperature 26.7 ± 3.2 °C humidity 36.4 ± 14.4% (O-257; DRETECKOREA Co., Seoul, Korea), and illuminance 10,327.9 ± 7986.2 Lux.

### 2.3. Experimental Protocol

In this study, olfactory stimulation was performed using 10 types of oils (lavender, rosemary, rose, eucalyptus, jasmine, geranium, chamomile, clary sage, thyme, and peppermint; Beleza Co., Seoul, Korea; extracted by pressing and distillation) to investigate the effect of olfactory stimulation on brainwave changes in the prefrontal cortex.

Aroma oils (0.85 µL, the concentration was set by referring to the treatment method of previous studies [26]) were injected using a micro-cylinder (80330; Hamilton; Hwashin Instrument Co., Seoul, Korea) into a three-hole joint flask (FJ1140-3000D; LABDIA; Seoul, Korea) and was dropped on a filter in the flask and diffused for 10 min. The piston pump (10RNS; G&M Tech Inc.; Yongin, Korea) was then operated and injected along the hose at a flow rate of 3 L/min, and the participants were instructed to breathe naturally through the attached glass funnels under their noses at intervals of 10 cm with their eyes closed (Figure 1).

Olfactory stimulation with the 10 aroma oils was randomized, and EEG was measured for 90 s for each treatment. During the EEG measurement, the participants were instructed to restrain extraneous movement and refrain from speaking. After olfactory stimulation with each oil, the blood pressure and pulse rate were measured. The participant rested for 30 s in the same place, and then the experiment was repeated in the same way. The duration of the experiment per participant was approximately 90 min.

### 2.4. Measurements

In this study, the EEG readings were analyzed to measure the psychophysiological indicators of female adults according to olfactory stimulation with the aromatic oils. The EEG analysis device used in this study was a wireless EEG measuring device (Quick-20, Cognionics, Inc., San Diego, CA, USA). The dry electrode of the wireless EEG device used in this study minimized the risk of electrical stimulation.

Electroencephalography (EEG) refers to the recording of electrical signals from the human brain at the scalp level [27]. This method records electrophysiological signals generated by brain activity by attaching a sensor to the surface of the scalp. The electrical signals mentioned correspond to the following frequency bands: delta (0–4 Hz), theta (4–8 Hz), alpha (8–12 Hz) and beta (12–30 Hz). Human actions, thoughts, and emotions can alter the brain wave activity at different frequencies. Alpha and beta waves are believed to be most closely related to human emotions. Alpha waves are known to correlate with reduced mental stress, increased relaxation, and improved memory, while beta waves correspond to clear and fast thinking [28,29].

The reference electrode was attached to the left ear cortex (A1). According to the international electrode method, this study performed EEG monitoring at Fp1 (left prefrontal cortex) and Fp2 (right prefrontal cortex; Figure 2).

Blood pressure and pulse rate were measured to assess the physiological changes after inhaling the fragrance of each oil. The physiological data included pulse rate (HRV), systolic blood pressure (mmHg), and diastolic blood pressure (mmHg). The blood pressure was measured using a blood pressure monitor (T4 with Intellisense; Omron Co., Kyoto, Japan) on the left arm of the participants at rest and after treatment with the 10 oils.

### 2.5. Data Analysis

The measured EEG data were analyzed using the Bio-scan (Bio-Tech, Daejeon, Korea) program. Because the difference in results between before and after stimulation was greatest at 30 s after stimulus presentation [31], data in the middle 30 s were extracted and analyzed.

Data were collected using a brain mapping program (Bioteck Analysis Software, Daejeon, Korea) to average EEG measurements during the experiment.

Data were collected by amplifying the electrical signal measured by attaching a dry electrode to the scalp. This device was designed to be harmless to humans and was certified by the European Commission and the Federal Communications Commission [32]. The electrode placement complied with the international 10–20 electrode arrangement system [18].

The collected EEG raw data were analyzed using power spectrum analysis to identify the relative alpha (RA) power spectrum, relative low beta (RLB) power spectrum, and the ratio of alpha to high beta (RAHB) [33] (Table 2).

The processed EEG data, pulse rate data, and blood pressure data were analyzed using the paired *t*-test. The analysis was performed using the SPSS (Version 25 for Windows; IBM, Armonk, NY, USA) program. Significance was set at *p* < 0.05. To analyze demographic information, descriptive statistics were performed on the mean, standard deviation, and percentage of each collection item using Microsoft Excel (Office 2007; Microsoft Corp., Redmond, WA, USA).

## 3. Results

### 3.1. Demographic Information

The average age of participants in this study was 31.13 ± 11.42 years, and 23 female adults participated (Table 3). The average body weight was 56.51 ± 7.96 kg. The average body mass index (BMI) was 21.43 ± 3.08 kg m^−2^, which was within the normal range according to the WHO standards.

As for the self-measured olfactory function of the participants, scent survey for screening (SSS) scored an average of 84.5 ± 11.9, and the visual analog scale (VAS) scored an average of 8.6 ± 1.2, which belonged to the normal olfactory group (Table 3).

### 3.2. EEG Responses of 10 Types of Aroma Oil

As a result of EEG changes in the prefrontal cortex of female adults upon olfactory stimulation with 10 oils, relaxation and comfort were activated, indicating physiological stability (*p* < 0.05; Table 4). The results of analyzing differences before and after olfactory stimulation with lavender oil are as follows: RA significantly increased in the left and right prefrontal cortex (*p* < 0.01; Table 4), and RLB significantly increased in the right prefrontal cortex (*p* < 0.05). Upon olfactory stimulation with rosemary oil, RA significantly increased during olfactory stimulation in the left and right prefrontal cortex (*p* < 0.05; Table 4), and RLB and RAHB were significantly increased in the right prefrontal cortex (*p* < 0.05). As a result of analyzing the brain waves before and after the olfactory stimulation with jasmine oil and clary sage oil, RA and RAHB were significantly increased in both hemispheres of the prefrontal cortex (*p* < 0.01; Table 4). Upon olfactory stimulation with rose oil, RLB was significantly increased in the right prefrontal cortex (*p* < 0.01; Table 4). Similarly, upon olfactory stimulation with geranium oil, RLB was significantly increased in both hemispheres of the prefrontal cortex (*p* < 0.01; Table 4). Olfactory stimulation with eucalyptus, chamomile, and thyme oil significantly increased RA. Eucalyptus and chamomile oil showed significant changes in the right prefrontal cortex (*p* < 0.05; Table 4), and significant changes were confirmed on both hemispheres of the prefrontal cortex during olfactory stimulation with thyme oil (*p* < 0.05; Table 4). As a result of analyzing the difference before and during olfactory stimulation with peppermint oil, RAHB was significantly increased in the right prefrontal cortex (*p* < 0.05; Table 4).

### 3.3. Pulse Rate and Blood Pressure Response of 10 Types of Aroma Oil

As a result of comparing the pulse rate and blood pressure before and after olfactory stimulation with 10 aroma oils, the difference between the pulse rate and diastolic blood pressure was not significant (Table 5). On the other hand, the difference in systolic blood pressure decreased overall, and all oils except rosemary and jasmine significantly decreased systolic blood pressure (Table 5).

## 4. Discussion

This study investigated the psychophysiological responses of olfactory stimulation with aroma oil, such as stabilization and relaxation of the prefrontal cortex, brain activity, and reduction of stress, on 23 female adults. Participants inhaled the scents of 10 types of aroma oil for 90 s each, and EEG readings, blood pressure, and pulse rate were measured to detect psychophysiological changes.

EEG of the prefrontal cortex upon olfactory stimulation with aroma oil showed significant increases in RA and RAHB, which are indicators of physiological stability and brain relaxation, and significant increase in RLB, an indicator of brain activity. Changes in EEG upon olfactory stimulation indicate that states of relaxation and concentration can occur at the same time to some extent [34].

Min et al. [35] found that the brains of perfumery researchers respond to odors primarily in the frontal lobe region and exhibit functions in the prefrontal or prefrontal cortex due to the occupational requirement of distinguishing or discriminating odors. Additionally, Watanuki et al. [36] found that beta wave activity was increased in the left frontal brain region when pleasant scents were smelled.

Alpha waves observed in the prefrontal cortex are related to the semi-awake state of REM sleep, meditation, and the inner state of peace and stillness [37]. Beta waves from the frontal cortex mainly indicate cognitive performance such as evaluation and decision of stimuli [38], and the relative low beta (RLB), an index analyzed in this study, is mainly activated in a stress-free state of concentration, work in a relaxed state, and learning. Jiang et al. [34] revealed that the average alpha wave value was higher in the experimental group than in the control group when the scent of the *Primula* plant was inhaled. Kim et al. [39] found that alpha waves tended to increase dramatically after participants sniffed fragrant extracts of orchid petals.

EEG of the prefrontal cortex following olfactory stimulation with aroma oil showed the following results. Olfactory stimulation with lavender oil significantly increased RA and RLB, indicating increased activity of alpha and beta waves. As a result of aromatherapy using lavender oil, the effect of lavender oil on stabilization and relaxation was confirmed compared with the results of previous studies in which beta waves in the frontal cortex increased [11] and alpha waves in the temporal cortex were significantly increased [14]. Further, when considering the effect of lavender scent on the brain waves of adult women with sleep disorders and in healthy women with good sleep quality, theta waves in the frontal cortex and beta waves in the occipital cortex increased. It was found that theta waves, implicated in sleep elevation, increased in all areas of the cerebral cortex [24]. A follow-up study is needed to examine the stimulation effect of aroma oil on sleep quality, cognitive function, and concentration, the results of which can be compared with the results of previous studies and the current study.

RA, RLB, and RAHB were significantly increased during olfactory stimulation with rosemary oil, which is consistent with previous findings [40], in which beta waves were increased in the anterior region of the brain. The findings of the current study indicate that activation can help improve concentration and memory.

Upon olfactory stimulation with eucalyptus, chamomile, and thyme oil, RA significantly increased in the prefrontal cortex. In the case of the scent stimulation with eucalyptus, the alpha wave activity of the cerebral cortex was significantly increased [41] compared to the previous study in which the alpha wave of the parietal cortex increased. In addition, various extracts of chamomile are well known for their anxiolytic and sedative properties [42]. In this study, the physiological effects of the main active ingredients were identified through the component analysis of aromatic oils. According to this finding it is necessary to examine the effects on the human body through analysis of the components of aromatic oils and the volatile organic compounds (VOC) in flower plants, which are the most common raw materials.

Olfactory stimulation with jasmine and clary sage oil resulted in significant increase in RA and RAHB in both hemispheres of the prefrontal cortex, which indicate brain stability and relaxation, and peppermint oil significantly increased RAHB in the right prefrontal cortex. Compared with the results of previous studies that showed an increase in beta waves in the center of the frontal cortex and the left occipital cortex upon olfactory stimulation with jasmine oil [43], the RAHB index shown in this study not only increased relative alpha waves, but also increased the ratio of beta waves. It can be seen that the results are similar to the results of previous studies. Further, the results of this study are similar to those of the study by Hassan et al. [20] on emotional changes induced by environmental contact, including horticultural activities. These studies consistently reported higher mean alpha and beta values, reflecting higher levels of relaxation and attention than a relatively steady state.

In this study, EEG changes through olfactory stimulation of aroma oil were measured using a wireless EEG device, but in a previous study, changes in the autonomic nervous system were revealed by measuring the concentration of oxyhemoglobin in the prefrontal cortex using near-infrared spectroscopy (NIRS) [17]. In the study, the olfactory stimulation of rose oil and orange oil decreased the concentration of oxyhemoglobin in the prefrontal cortex, leading to psychophysiological stability and relaxation.

As a result of this study, the RLB of the right frontal lobe was significantly increased according to the olfactory stimulation of rose oil, which signifies the ability to show concentration without tension and stress, as a beta wave with a slow frequency among beta waves. It was confirmed that the decrease in oxyhemoglobin concentration during the olfactory stimulation of rose oil, similar to that revealed in the previous study, indicates a state of being able to concentrate while resting.

In order to support the autonomic nervous system and psychophysiological responses of olfactory stimulation, follow-up studies are needed to analyze the psychophysiological changes of olfactory stimulation using aroma oil and various wearable devices.

In addition, EEG changes were analyzed focusing on the prefrontal cortex, which is related to cognition, emotion, and mental function. It is also considered necessary to study physiological changes according to the influx of the human body, such as blending two or more types or application to the skin rather than inhalation.

Notably, systolic blood pressure decreased significantly after olfactory stimulation with aroma oil. As shown in the EEG results, it is judged that the increase in blood pressure is reduced due to physiological stability and stress reduction; this finding is similar to that of a previous study in which inhalation of lavender, marjoram, and clary sage oils significantly reduced stress and systolic blood pressure [44]. Previous studies have shown that the sympathetic and parasympathetic nervous systems produce changes in blood pressure and heart rate when emotions are affected by stress or relaxation [45,46]. A stressful state induces an increase in blood pressure and pulse rate, and a relaxed emotional state induces a decrease [47,48].

Therefore, based on the changes in EEG and systolic blood pressure analyzed in this study, olfactory stimulation with aroma oil is effective in reducing stress and in psychophysiological relaxation.

## 5. Conclusions

This study was conducted to measure the effect of olfactory stimulation with aroma oil on the physiological response of female adults. Stability and relaxation of the prefrontal cortex, increased brain activity, and decreased systolic blood pressure showed a positive effect on reduced mental stress.

In this study, the psychophysiological effects of aroma oil were analyzed in female adults, and future research is needed for a scientific approach to horticultural treatment by expanding the scope of application according to sex, age, and disease. In addition, it is expected that the results of this study will be useful basic data for the selection of suitable plants, according to the characteristics and goals of the subject and the composition of the program when composing a horticultural treatment program in the future.

## Figures and Tables

**Figure 1 ijerph-19-05196-f001:**
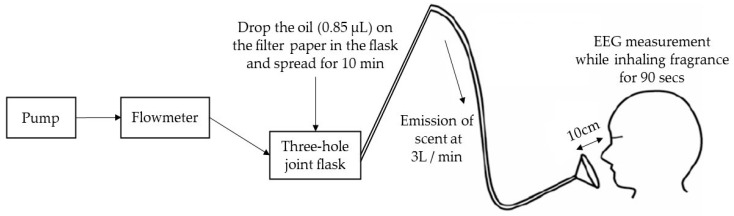
Experimental protocol; EEG: electroencephalogram.

**Figure 2 ijerph-19-05196-f002:**
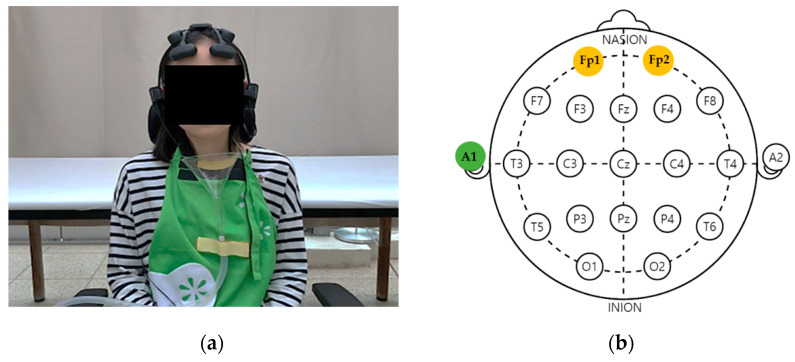
Wearing instrument: (**a**) wireless dry electroencephalography device (Quick-20, Cognionics, Inc., San Diego, CA, USA); (**b**) international electrode arrangement [30].

**Table 1 ijerph-19-05196-t001:** Criteria and cautions for selecting/excluding research participants.

Selection Criteria	A person who does not have psychopathological diseases and does not take related drugs A person who is right-hand dominant
Exclusion criteria	A person who does not agree to participate in the research even though she fully understands the contents of the research A person with a history of cardiovascular diseases such as high blood pressure, unstable angina, heart attack, and heart surgery A person with diseases such as olfactory function-related disorders, allergies, respiratory diseases, and insomnia A person who is pregnant, lactating, or menstruating
Requirements of participation	Stop drinking the day before the experiment Prohibit excessive physical activity (e.g., breathless high intensity physical activity for more than 60 min) on the day before the experiment Prohibit consumption of caffeinated beverages and smoking within 3 h before the experiment Do not use cosmetics with strong scents such as perfumes and sprays on the day of the experiment

**Table 2 ijerph-19-05196-t002:** EEG power spectrum indicators used in the study.

Analysis Indicators	The Full Name of the EEG Power Spectrum Indicator	Wavelength Range (Hz)
RA	Relative alpha power spectrum	(8–13)/(4–50)
RLB	Relative low beta power spectrum	(12–15)/(4–50)
RAHB	Ratio of alpha to high beta	(8–13)/(20–30)

**Table 3 ijerph-19-05196-t003:** Descriptive characteristics of the participants (N = 23).

Variable	
Sex	% (N)
Female	100 (23)
	Mean (SD ^1^)
Age (years)	31.1 (11.4)
Height (cm)	162.5 (6.41)
Body weight (kg)	56.5 (8.0)
Body mass index (kg·m^−2^) ^2^	21.4 (3.1)
SSS ^3^	84.5 (11.9)
VAS ^4^	8.6 (1.2)

^1^ Standard Deviation. ^2^ Body mass index = weight (kg)/height (m^2^). ^3^ SSS = Scent survey for screening; a score of 47 or higher is a normal olfactory group. ^4^ VAS = Visual analog scale; A score of 5 or higher is a normal olfactory group.

**Table 4 ijerph-19-05196-t004:** Results of relative alpha power spectrum (RA), relative low beta power spectrum (RLB) and ratio of alpha to high beta (RAHB) using electroencephalography due to olfactory stimulation with 10 types of aroma oil.

		RA ^1^		RLB ^2^		RAHB ^3^	
		Fp1	Fp2	Fp1	Fp2	Fp1	Fp2
		Mean ± SD ^4^		Mean ± SD		Mean ± SD	
Lavender oil	Resting	0.222 ± 0.060	0.217 ± 0.061	0.067 ± 0.012	0.066 ± 0.012	1.582 ± 0.559	1.530 ± 0.574
	Treatment	0.271 ± 0.083	0.270 ± 0.086	0.072 ± 0.008	0.074 ± 0.009	1.925 ± 0.820	1.902 ± 0.824
	Significance ^5^	0.009 **	0.009 **	NS	0.026 *	NS	NS
Rosemary oil	Resting	0.222 ± 0.060	0.217 ± 0.061	0.067 ± 0.012	0.066 ± 0.012	1.582 ± 0.559	1.530 ± 0.574
	Treatment	0.257 ± 0.081	0.256 ± 0.080	0.074 ± 0.016	0.074 ± 0.013	1.891 ± 0.791	1.916 ± 0.773
	Significance	0.047 *	0.037 *	NS	0.015 *	NS	0.036 *
Rose oil	Resting	0.222 ± 0.060	0.217 ± 0.061	0.067 ± 0.012	0.066 ± 0.012	1.582 ± 0.559	1.530 ± 0.574
	Treatment	0.248 ± 0.073	0.248 ± 0.075	0.073 ± 0.012	0.074 ± 0.012	1.723 ± 0.689	1.708 ± 0.705
	Significance	NS	NS	NS	0.005 **	NS	NS
Eucalyptus oil	Resting	0.222 ± 0.060	0.217 ± 0.061	0.067 ± 0.012	0.066 ± 0.012	1.582 ± 0.559	1.530 ± 0.574
	Treatment	0.257 ± 0.076	0.257 ± 0.082	0.071 ± 0.012	0.072 ± 0.012	1.847 ± 0.668	1.815 ± 0.724
	Significance	NS	0.044 *	NS	NS	NS	NS
Jasmine oil	Resting	0.222 ± 0.060	0.217 ± 0.061	0.067 ± 0.012	0.066 ± 0.012	1.582 ± 0.559	1.530 ± 0.574
	Treatment	0.273 ± 0.085	0.271 ± 0.087	0.067 ± 0.011	0.068 ± 0.011	2.012 ± 0.709	1.975 ± 0.733
	Significance	0.033 *	0.026 *	NS	NS	0.039 *	0.029 *
Geranium oil	Resting	0.222 ± 0.060	0.217 ± 0.061	0.067 ± 0.012	0.066 ± 0.012	1.582 ± 0.559	1.530 ± 0.574
	Treatment	0.254 ± 0.074	0.256 ± 0.074	0.073 ± 0.012	0.074 ± 0.011	1.762 ± 0.731	1.787 ± 0.731
	Significance	NS	NS	0.049 *	0.009 **	NS	NS
Chamomile oil	Resting	0.222 ± 0.060	0.217 ± 0.061	0.067 ± 0.012	0.066 ± 0.012	1.582 ± 0.559	1.530 ± 0.574
	Treatment	0.257 ± 0.082	0.262 ± 0.081	0.071 ± 0.011	0.071 ± 0.009	1.890 ± 0.776	1.868 ± 0.787
	Significance	NS	0.016 *	NS	NS	NS	NS
Clary sage oil	Resting	0.222 ± 0.060	0.217 ± 0.061	0.067 ± 0.012	0.066 ± 0.012	1.582 ± 0.559	1.530 ± 0.574
	Treatment	0.269 ± 0.071	0.254 ± 0.069	0.073 ± 0.014	0.072 ± 0.012	2.062 ± 0.743	1.954 ± 0.896
	Significance	0.009 **	0.045 *	NS	NS	0.009 **	0.039 *
Thyme oil	Resting	0.222 ± 0.060	0.217 ± 0.061	0.067 ± 0.012	0.066 ± 0.012	1.582 ± 0.559	1.530 ± 0.574
	Treatment	0.258 ± 0.082	0.260 ± 0.084	0.074 ± 0.018	0.074 ± 0.017	1.891 ± 0.837	1.896 ± 0.839
	Significance	0.034 *	0.019 *	NS	NS	NS	NS
Peppermint oil	Resting	0.222 ± 0.060	0.217 ± 0.061	0.067 ± 0.012	0.066 ± 0.012	1.582 ± 0.559	1.530 ± 0.574
	Treatment	0.250 ± 0.078	0.250 ± 0.081	0.070 ± 0.015	0.070 ± 0.016	1.879 ± 0.677	1.919 ± 0.696
	Significance	NS	NS	NS	NS	NS	0.047 *

^1^ RA was calculated by [alpha (8–13 Hz) power]/[total frequency (4–50 Hz) power]. ^2^ RLB was calculated by [low beta (12–15 Hz) power]/[total frequency (4–50 Hz) power]. ^3^ RAHB was calculated by [alpha (8–13 Hz) power]/[high beta (20–30 Hz) power]. ^4^ SD: standard deviation. ^5^ * *p* < 0.05, ** < 0.01 using the Paired *t*-test.

**Table 5 ijerph-19-05196-t005:** Change in pulse rate due to olfactory stimulation with 10 types of aroma oil.

	Before	Lavender	Rosemary	Rose	Eucalyptus	Jasmine	Geranium	Chamomile	Clary Sage	Thyme	Peppermint
	Mean ± SD ^1^										
Pulse rate	72.26 ± 9.55	72.96 ± 9.83	73.04 ± 8.88	72.61 ± 8.40	72.04 ± 9.09	73.61 ± 9.60	71.52 ± 7.98	72.65 ± 9.27	73.22 ± 8.88	72.09 ± 9.14	72.26 ± 9.01
Significance ^2^		0.471	0.430	0.767	0.848	0.151	0.475	0.716	0.426	0.862	1.000
Systolic pressure	109.00 ± 10.46	105.39 ± 9.88	104.70 ± 9.73	106.26 ± 10.89	104.17 ± 9.65	106.91 ± 11.05	104.74 ± 10.72	104.22 ± 10.24	106.17 ± 11.37	103.91 ± 10.57	104.74 ± 10.23
Significance		0.003 **	0.001 ***	0.058	0.002 **	0.062	0.001 ***	0.000 ***	0.033 *	0.000 ***	0.008 **
Diastolic pressure	69.91 ± 8.16	69.09 ± 6.50	69.00 ± 7.01	69.70 ± 7.10	69.78 ± 7.42	69.26 ± 6.86	69.09 ± 8.27	69.57 ± 6.69	71.57 ± 6.29	68.57 ± 8.70	69.43 ± 6.56
Significance		0.382	0.423	0.847	0.916	0.495	0.436	0.755	0.120	0.182	0.709

^1.^ SD: standard deviation. ^2^ * *p* < 0.05, ** < 0.01, *** < 0.001 using the paired *t*-test.

## Data Availability

The datasets generated for this study are available on request to the corresponding author.

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
