# Peer review of "Effects of Olfactory Stimulation with Aroma Oils on Psychophysiological Responses of Female Adults"

_ijerph, 2022, doi:10.3390/ijerph19095196_

Round 1

Reviewer 1 Report

The paper presents an interesting study about the use of aromatherapy and its physiological effects in a cohort of adult women.

Overall, the manuscript is pleasant to read and does not present particular drawbacks in terms of methodology, overall coherence and scientific value. Therefore, I only have a couple minor concerns to be addressed before deeming it publishable for this journal.

Concerning the state of the art, I would also include some studies evaluating the effects of odors and olfactory stimulation in the autonomic nervous system by means of wearable sensors. Such articles could also serve as a reference for the discussuion part, to compare the results obtained here with those coming from such studies.

From a methodological point of view, why not including also the study of heart rate variability in your investigation? Please, discuss.

Overall, please, check for typos throughout the manuscript.

Author Response

Dear Reviewer,

We appreciate the reviewer’s thoughtful suggestions and recommendations. We have integrated these inputs into the revised manuscript; the revised contents in response to the reviewer’s comments are blue in text.

We have integrated these inputs into the revised manuscript. First, among the previous studies mentioned in the introduction (Igarashi et al., 2014), a study on the olfactory stimulation of aroma oil using a wearable device called near-infrared spectroscopy (NIRS) was added to discussion and compared with the results of this study. In addition, it was revealed that it is necessary to prove the psychophysiological effect of olfactory stimulation of aroma oil with a device that can measure changes in the autonomic nervous system in addition to the brain wave device in the future.

The investigation method for ECG variability was conducted during EEG measurement, but statistical processing did not show a significant difference. Therefore, the methodology and results of the electrocardiogram were not described in this research paper.

Also, we checked for typos in sentences and tables. Thank you.

Reviewer 2 Report

The paper “Effects of Olfactory Stimulation with Aroma Oils on Psychophysiological Responses of Female Adults” is a well conducted empirical study about aroma oils and neuroscience. The authors have study 23 women that used aroma oils performing a set of electrophysiologic measurements, together with blood pressure and heart-rate records. The work was focused on the e prefrontal cortex brain activity. In fact, the study compare before and during the inhalation of aroma oil. The quantities used to compare the signal in the Fp1 e Fp2 were: the Relative alpha power spectrum, the Relative low beta power spectrum and the Ratio of alpha to high beta. All these quantities were well defined, the statistical treatment was also adequate. The experimental methodology, since the criteria for selecting/excluding research participants to the experimental protocol of the measurements, were also well described. The results of the paper were clearly exposed.

I recommend the publication of the paper with small modifications. Only two minor points that could enhance the work called may attention, they are enumerated bellow:

1) The first paragraph of the section Results, which concern the demographic information, could be transferred to the methodology section.

2) Tables 4 to 13 could be joined into a single table. This is a suggestion, I think it would be better to the reader to visualize all information in a single table, including the non-significant results.

Author Response

Dear Reviewer,

We appreciate the reviewer’s thoughtful suggestions and recommendations. We have integrated these inputs into the revised manuscript; the revised contents in response to the reviewer’s comments are blue in text.

Since the content of demographic information is the result of the characteristics of the participants who were recruited by the recruitment method presented in the research method. So it is thought that it is correct to describe it in the results section. However, according to the reviewer's advice, the previous manuscript has already stated the average age in the recruitment method, and the resulting description was deleted and corrected.

Also, Tables 4 to 13 are combined into one table (Table 4). It has been modified to make it easier to understand the brain wave analysis results of 10 types of aroma oil, including even insignificant result data. Thank you.

Reviewer 3 Report

Dear Authors,

You have total 14 tables in your revised manuscript. That's good in the sense that you have enough data or outcomes to discuss from your study. 

BUT, when I see the 'Results' section, to me, these tables with lots of information are not properly explained. They are scientifically explained in the manuscript to make attention to the readers.

I will suggest taking time, taking one table and explaining it. Then take the next table - explain it - and if related then connect with previous or/and next table. This will make the 'Results' section more organized than the present shape, and the readers will feel the interest to continue the reading.

I would like to say here that we don't write manuscripts only for the expert parole but also for young researchers who are new in the area where they get interested.

Thanks

Reviewer

Author Response

Dear Reviewer,

We appreciate the reviewer’s thoughtful suggestions and recommendations. We have integrated these inputs into the revised manuscript.

We agree with the point you pointed out that the listing of tables and result descriptions may be difficult for the reader to understand. Therefore, the explanation of the EEG results of 10 types of aroma oil was easily interpreted in one paragraph. After that, I added a table that combines Tables 4-13 into one (Table 4), resulting in a better visualization. Thank you.